# Bilateral ankle dorsiflexion force control impairments in older adults

**Do-Kyung Ko**[1,2], **Hanall Lee**[1,2], **Hajun Lee**[1,2], **Nyeonju Kang**[1,2,3]*

**1** Department of Human Movement Science, Incheon National University, Incheon, South Korea, **2** Neuromechanical Rehabilitation Research Laboratory, Incheon National University, Incheon, South Korea, **3** Division of Sport Science, Sport Science Institute and Health Promotion Center, Incheon National University, Incheon, South Korea

* nyunju@inu.ac.kr

## Abstract

Age-related impairments in ankle dorsiflexion force modulation are associated with gait and balance control deficits and greater fall risk in older adults. This study aimed to investigate age-related changes in bilateral ankle dorsiflexion force control capabilities compared with those for younger adults. The study enrolled 25 older and 25 younger adults. They performed bilateral ankle dorsiflexion force control at 10% and 40% of maximum voluntary contraction (MVC), for vision and no-vision conditions, respectively. Bilateral force control performances were evaluated by calculating force accuracy, variability, and complexity. To estimate bilateral force coordination between feet, vector coding and uncontrolled manifold variables were quantified. Additional correlation analyses were performed to determine potential relationships between age and force control variables in older adults. Older adults demonstrated significantly lower force accuracy with greater overshooting at 10% of MVC than those for younger adults. At 10% and 40% of MVC, older adults significantly showed more variable and less complex force outputs, and these patterns appeared in both vision and no-vision conditions. Moreover, older adults revealed significantly less anti-phase force coordination patterns and lower bilateral motor synergies with increased bad variability than younger adults. The correlation analyses found that lower complexity of bilateral forces was significantly related to increased age. These findings suggest that aging may impair sensorimotor control capabilities in the lower extremities. Considering the importance of ankle dorsiflexion for executing many activities of daily living, future studies may focus on developing training programs for advancing bilateral ankle dorsiflexion force control capabilities.

## Introduction

Aging induces progressive impairments in the motor system, which increase the prevalence of functional disabilities in the lower extremities such as slower walking speed [1–3]. These motor deficits interfere with independent living in older people [4]. Age-related ankle joint function impairments can induce a loss of static and dynamic balance control [5,6], further previous studies have suggested that strength and fine motor control during ankle dorsiflexion are crucial to minimize the risk of falls in the aging population [7–10]. Thus, estimating

**Data availability statement:** All relevant data are within the paper and its Supporting information files.

**Funding:** This work was supported by Incheon National University Research Grant in 2022 (2022-0094) awarded to NK. The funders had no role in study design, data collection and analysis, decision to publish, or preparation of the manuscript.

**Competing interests:** The authors have declared that no competing interests exist.

altered ankle dorsiflexion functions may effectively quantify the progression of neuromuscular deficits in the lower extremities of older adults.

Previous studies have indicated that aging typically reduced flexibility (e.g., less range of ankle dorsiflexion motion), impaired mobility (e.g., delayed ankle dorsiflexion during the gait swing phase), and caused muscle weakness (e.g., lower strength in the tibialis anterior muscle) of the lower extremities [8,11–13]. Moreover, many studies have used an isometric force control paradigm to assess age-related changes in fine motor control functions during ankle dorsiflexion while processing online visual information [14–16]. Given that age-related changes in the central and peripheral nervous systems may affect the ability to simultaneously correct and maintain isometric force outputs near a targeted level [17,18], isometric ankle force control capabilities were impaired with aging [19–21]. The older group showed greater force error and variability at lower targeted force levels (e.g., 5%–10% of maximal voluntary contraction [MVC]) because of altered visuomotor networks in the brain and activation of motor neuron pools [20,21]. However, these ankle dorsiflexion force control impairments were mostly unilateral (e.g., dominant leg). In fact, age-induced movement control deficits were frequently observed in walking, postural control, and sit-to-stand that require bilateral actions in the lower extremities [22]. Given that many daily activities for living independent life consist of fundamental motor skills related to interlimb coordination between feet [23,24], investigating bilateral force control and coordination in the lower extremities is necessary for understanding age-induced functional impairments. Studies on bilateral ankle dorsiflexion force control capabilities in older adults may provide additional information on how aging influences cooperative fine motor control between feet. Further, these findings can be used for developing new rehabilitation programs (e.g., robotic exoskeletons for improving lower limb coordination) targeting functional recovery of bilateral ankle movements in aging population [25].

Bilateral force control capabilities in the lower extremities can be estimated by quantifying the accuracy, variability, and complexity (i.e., temporal structure of variability) of the total summed forces produced by the left and right sides of extremities [26–29]. Greater force accuracy, decreased force variability, and higher complexity normally indicated better bilateral force control capabilities affected by organism, environment, and task constraints (e.g., different populations, visual feedback, and targeted force level conditions) [30]. Moreover, estimating interlimb force coordination patterns can reveal the performer's motor control strategies that require more complex neural involvements to optimize bilateral force control performances [30–32]. Interlimb force coordination strategies have two aspects to consider: (1) coordination pattern within-trial analysis and (2) coordination pattern between-trial analysis. A former approach can estimate the strength of interlimb force coordination according to in-phase and anti-phase actions between feet so that more anti-phasic force coordination within a trial predominantly appeared with better bilateral force control performances [32,33]. A latter approach can assess how a performer successfully coordinates interlimb motor actions synergistically across repetitive trials [34,35]. The uncontrolled manifold (UCM) analysis is a representative between-trial approach, and an increased index of bilateral motor synergies (i.e., a ratio of good variability relative to bad variability) from the UCM approach denotes better interlimb force coordination across multiple trials, which contribute to overall task stabilization indicating improvements across multiple force control trials [34,35].

This study aimed to examine bilateral ankle dorsiflexion force control and coordination patterns in older adults. To the best of our knowledge, this is the first study to investigate bilateral ankle dorsiflexion force control capabilities in aging population. Both older and younger participants executed isometric force control tasks across two experimental conditions: (1) visual feedback (vision vs. no-vision) and (2) targeted force level (10% and 40% of MVC).

Force accuracy, variability, and complexity were measured, and interlimb force coordination patterns between feet were determined using within-trial and between-trial approaches. Considering the potential age-induced progressive degeneration in lower limb functions [3], we hypothesized that older adults would reveal a lower accurate, more variable, and less complex ankle dorsiflexion forces produced by feet while generating impaired interlimb force coordination patterns within a trial and across multiple trials than younger adults.

## Materials and methods

### Participants

Participants were recruited between January 2, 2019 and February 15, 2024. The study enrolled 25 older adults (age: $M \pm SD$ = 64.9 ± 3.8 years; physical activity: $M \pm SD$ = 222.8 ± 105.2 minute/weak; 15 females) and 25 younger adults (age: $M \pm SD$ = 23.0 ± 2.4 years; physical activity: $M \pm SD$ = 283.6 ± 182.1 minute/weak; 15 females). All subject had no cognitive impairments, vision disorders, and musculoskeletal deficits in their lower extremities. The dominant leg of all participants was the right leg based on the ball-kicking test [36,37]. Table 1 shows specific demographic details. Before starting the experiment, all participants read the study protocols and signed informed consent approved by the University's Institutional Review Board (No. 7007971-201810-002A).

### Experimental procedures

A customized isometric foot-force measurement system (SEED TECH Co., Ttd., Bucheon, South Korea) was used for executing bilateral ankle dorsiflexion force control tasks. Participants sat 80 cm in front of a 54.6 cm LED screen and placed their feet on the customized platforms. Then, we directly adjusted the chair and instructed them to maintain proper joint positions following joint angle ranges: 90°–95° of hip flexion, 90°–100° of knee flexion, and approximately 90° of ankle dorsiflexion (Fig 1a). During tasks, we continuously monitored and ensured that participants maintain proper joint positions. Two force transducers were embedded to the left and right sides of the platforms (Micro Load Cell-CZL635-3135, range = 330 lbs, Phidgets Inc., Calgary, Canada). Force signals were collected at 100 Hz of sampling rate using a 16-bit analog-to-digital converter (A/D; ADS1148 16-Bit 2kSPS; minimum detectable force = 0.0192 N) and amplified with an excitation voltage of 5 V using an INA122 (Texas Instruments Inc., Dallas, TX, USA). All experimental procedures were administered using a customized Microsoft Visual C++ program (Microsoft Corp., Redmond, WA, USA).

To minimize effects of different ankle dorsiflexion strength across participants on force control tasks, we normalized targeted force levels based on each participant's MVC including 10% and 40% MVC [38]. Initially, participants performed two MVC trials by producing

**Table 1. Demographic details for participants.**

| Characteristics | Older group | Younger group | P-value |
|---|---|---|---|
| Sample size | 25 | 25 | – |
| Sex (female: male) | 15: 10 | 15: 10 | – |
| Age (years) | 64.9 ± 3.8 | 23.0 ± 2.4 | $P < 0.001$* |
| Physical activity (minute/weak) | 222.8 ± 105.2 | 283.6 ± 182.1 | $P = 0.244$ |
| Leg dominance | 25 right leg | 25 right leg | – |
| Bimanual ankle dorsiflexion MVC (kg) | 40.6 ± 11.5 | 51.0 ± 14.4 | $P = 0.007$* |

MVC = maximal voluntary contraction. Data are given as mean ± standard deviation. Asterisk (*) indicates $P < 0.05$.

isometric ankle dorsiflexion forces by feet (trial duration = 3 s and 60 s of rest between trials), and a peak force value (i.e., the maximal value of the sum of the left and right foot forces) was identified for each trial. Then, 10% and 40% of the mean of two peak force values were calculated to set two submaximal targeted force levels. During bilateral ankle dorsiflexion force control tasks, participants tried to generate and match their sum of the left and right foot forces around a targeted force level for 20 s. Four different experimental blocks were randomly administered including a combination of two targeted force levels (i.e., 10% and 40% of MVCs) and two visual information conditions (i.e., vision and no-vision; Fig 1b). In the vision condition, participants received real-time visual information describing a targeted force level (i.e., white horizontal line) and the sum of bilateral forces (i.e., red trajectory line). In the no-vision condition, visual information displaying the sum of bilateral forces was unavailable after the initial 5 s. Before starting an experimental block, we provided one practice trial for familiarization. Participants completed ten consecutive trials for each experimental block. To minimize potential fatigue effects, a 30 s rest between trials and a 60 s rest between conditions were provided [33,39,40].

## Data analysis

Offline analyses were performed using the MatLab program (MathWorks TM Inc., Natick, USA). To minimize the initial adjustment and early termination effects, the initial 5 s and

**Fig 1. Experimental setup. (a)** Bilateral ankle force control task and **(b)** Two types of visual information: force production (red trajectory line) and targeted force level (white horizontal line).

the final 1 s of force signals for each trial were eliminated. The remaining 14 s of force signals were also low-pass filtered with a 30-Hz cutoff frequency using a bidirectional fourth-order Butterworth filter.

Bilateral force control performances were estimated by calculating force accuracy, variability, and complexity. We used relative variables, including relative root-mean-square error (rRMSE), relative bias error (rBE), and coefficient of variation (%CV), to minimize potential distortions by individual force levels [41]. Force accuracy was quantified using rRMSE (Eq 1) consistent with previous studies [42–44]. To specify the direction of force errors, rBE (Eq 2) was also estimated [45]. More positive values of rBE indicate overshooting errors, whereas more negative values mean undershooting errors relative to a targeted level.

$$rRMSE = \sqrt{\frac{\sum_{i=1}^{N}\left(F_i - Targeted\ force\right)^2}{N \times Targeted\ force}} \tag{1}$$

$$rBE = \frac{\sum_{i=1}^{N}(F_i - Targeted\ force)}{N \times Targeted\ force} \tag{2}$$

$F_i =$ bilateral force at sample $i$, $N =$ number of data samples

Force variability was estimated by quantifying the %CV ( $SD\,/\,mean\ force \times 100$ ) of the force outputs [17]. r force complexity, we used the refined composite multiscale sample entropy (rcMSE) consistent with the following procedures [46]: (1) create a coarse-grained time series using $k$ refer to nonoverlapping windows that range from 1 to the scale factor $\tau$, (2) use $m$ (number of similar vector lengths) and $r$ (tolerance range; usually $0.2 \times SD$ ) parameters, and calculate the number of matched vector pairs $n_{k,\tau}^{m+1}$ and $n_{k,\tau}^{m}$ in each coarse-grained time series, (3) the rcMSE value at a scale factor of $\tau$ was defined as the logarithm of the ratio of $\overline{n}_{k,\tau}^{m+1}$, to $\overline{n}_{k,\tau}^{m}$ (Eqs 3), and (4) the index of force complexity was calculated as the sum of rcMSE values with scales from 1 to $\tau_{Max}$ (Eq 4).

$$rcMSE\left(time\ series, \tau, m, r\right) = -\ln\left(\frac{\sum_{k=1}^{\tau} n_{k,\tau}^{m+1}}{\sum_{k=1}^{\tau} n_{k,\tau}^{m}}\right) \tag{3}$$

$$Complexity\ index = \sum_{\tau_{Max}}^{\tau=1} rcMSE\left(time\ series, \tau, m, r\right) \tag{4}$$

In this study, m to 2, r to 0.2 × SD of force data, and $\tau_{Max}$ to 10 were used. A higher complexity index means a more complex force control pattern.

Within-trial bilateral force coordination patterns in the lower extremities were evaluated using the vector coding approach that can estimate the frequency of coordination patterns between two segments [47]. In scatterplots that reflect the left and right foot force outputs, the direction of the trajectories over time is quantified by the coupling angle. Specifically, coupling angles are calculated using a vector for two successive data samples relative to the right transverse axis of the scatterplot (Eq 5; Fig 2a).

$$Coupling\ angle\ (\theta) = \tan^{-1}\left(\frac{RF_{i+1} - RF_i}{LF_{i+1} - LF_i}\right) \tag{5}$$

$LF_i =$ left foot force at sample $i$, $RF_i =$ right foot force at sample $i$

The coupling angle ranges from 0° to 360°, and coordination patterns were categorized as follows: (1) in-phase ($22.5° \leq \theta < 67.5°$, $202.5° \leq \theta < 247.5°$) and (2) anti-phase ($112.5° \leq \theta < 157.5°$, $292.5° \leq \theta < 337.5°$). More frequency in-phase patterns denote impaired interlimb force coordination, whereas greater frequency in anti-phase patterns indicates complementary coordination between feet [32,33].

Furthermore, the UCM analysis was conducted to estimate bilateral force coordination in the lower extremities across multiple trials [48]. First, the mean of the left and right foot forces for each trial was calculated, and the mean force values relative to a targeted force level (i.e., $100 \times$ left mean force / targeted force level and $100 \times$ right mean force / targeted force level) were normalized. For each trial, two normalized mean force values were considered as a pair of elemental variables, and the same calculations for ten trials were performed to generate ten pairs of elemental variables. Then, all pairs of elemental variables were projected to the UCM sub-space and orthogonal to the UCM sub-space (i.e., ORT sub-space). Good variability ($V_{UCM}$) is a variance of elemental variables projected to the UCM sub-space. Although altered levels of $V_{UCM}$ do not influence task performances, greater $V_{UCM}$ is considered a flexibility of interlimb coordination strategies (e.g., various motor solutions) across multiple trials. Bad variability ($V_{ORT}$) is the variance of elemental variables projected to the ORT sub-space, and increased levels of $V_{ORT}$ interfere with task stabilization leading to lower task performances (Fig 2b). Finally, bilateral motor synergies ($V_{Index}$), which is a ratio of $V_{UCM}$ to $V_{ORT}$ (Eq 6) were calculated, and $V_{Index}$ values were Z-transformed for additional parametric statistical analyses (Eq 7). Thus, greater values of $V_{Index}$ indicate improvements in bilateral force coordination patterns across multiple trials.

$$V_{Index} = \frac{V_{UCM} / df_{UCM} - V_{ORT} / df_{ORT}}{V_{TOT} / df_{TOT}} \qquad (6)$$

$$V_{TOT} = V_{UCM} + V_{ORT}$$

**a**

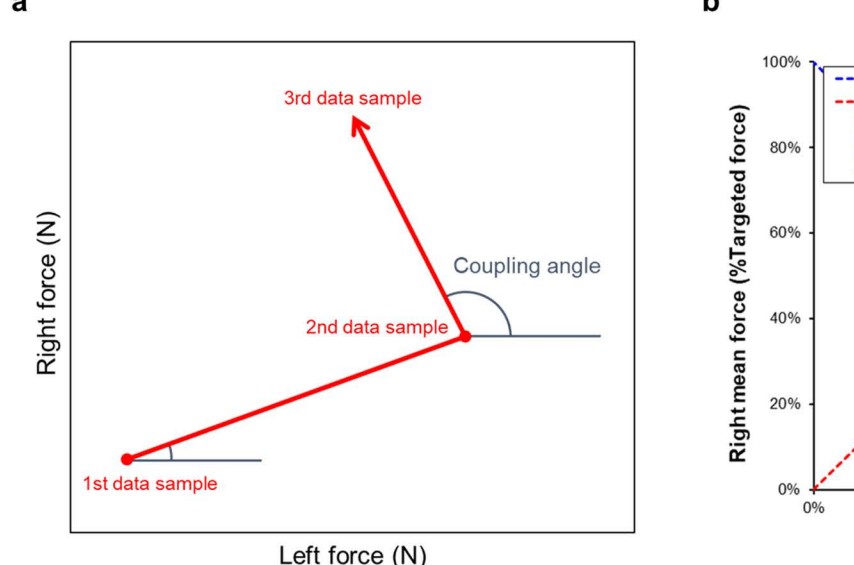

**b**

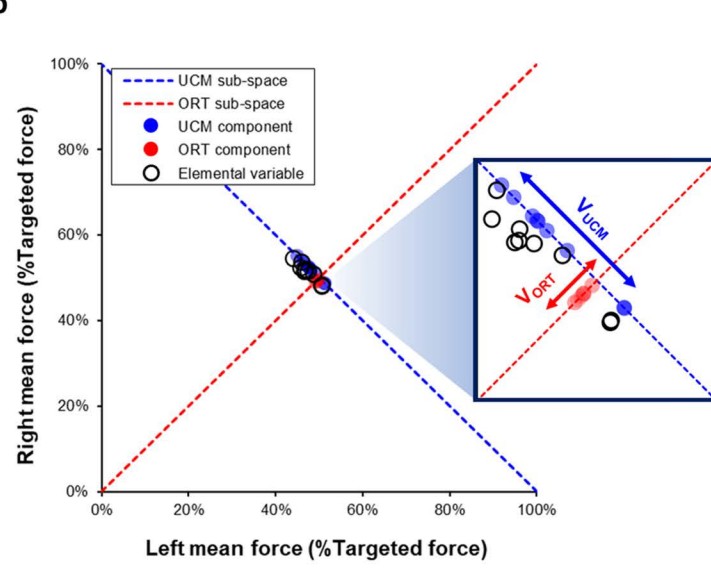

**Fig 2. Bilateral force coordination analyses.** (a) Coupling angle calculation and (b) Uncontrolled manifold analysis.

$$df = degrees\ of\ freedom\ \left(df_{UCM} = 1, df_{ORT} = 1,\ and\ df_{TOT} = 2\right)$$

$$V_{Index}\left(Z-transformed\right) = 0.5 \times\ \ln\frac{2+V_{Index}}{2-V_{Index}} \tag{7}$$

## Statistical analysis

The normality of distribution for all dependent variables was confirmed by the Shapiro Wilk tests. All dependent variables were analyzed by the three-way mixed model (Group × Force Level × Vision Condition; 2 × 2 × 2) analysis of variance (ANOVA) with repeated measures on the last two factors. For the post hoc analysis, Bonferroni's pairwise comparisons were used. For older adults, Pearson's correlation analyses were conducted to investigate the potential relationships between age and bilateral force control outcome measures in the lower extremities. IBM SPSS Statistics version 28 (IBM Corp., Armonk, NY, USA) was used for all statistical analyses, and the alpha level was set at 0.05.

## Results

### Bilateral force control performances

For force accuracy, three-way mixed ANOVA on the rRMSE revealed a significant Group × Force Level × Vision Condition interaction [$F(1, 48) = 9.013$; $P = 0.004$; partial $\eta^2 = 0.158$; Fig 3a]. Bonferroni's pairwise comparisons revealed that the older group produced higher rRMSE values than the younger group at 10% of MVC for vision ($P < 0.001$) and no-vision conditions ($P < 0.001$), respectively. The analysis of the rBE showed a significant Group × Force Level × Vision Condition interaction [$F(1, 48) = 5.755$; $P = 0.020$; partial $\eta^2 = 0.107$; Fig 3b]. In the post hoc analyses, the older group showed greater overshooting than the younger group at 10% of MVC for vision ($P = 0.007$) and no-vision conditions ($P < 0.001$), respectively. These results indicate that older adults revealed higher overshot force errors in bilateral lower extremities, and these patterns increased when visual feedback was unavailable.

The analysis of the %CV reported a significant Group × Force Level × Vision Condition interaction [$F(1, 48) = 5.304$; $P = 0.026$; partial $\eta^2 = 0.100$; Fig 3c]. In the post hoc analyses, the older group produced higher %CV values than the younger group at 10% of MVC in the vision ($P = 0.013$) and no-vision conditions ($P = 0.012$), respectively. At 40% of MVC, the older group showed higher %CV values than the younger group in the vision condition ($P = 0.045$). Moreover, the analysis of rcMSE reported a significant Group × Force Level × Vision Condition interaction [$F(1, 48) = 5.106$; $P = 0.028$; partial $\eta^2 = 0.096$; Fig 3d]. The older group showed lower rcMSE values than the younger group at 10% of MVC in the vision ($P = 0.036$) and no-vision conditions ($P = 0.008$), respectively. At 40% of MVC, the older group showed lower rcMSE values than the younger group in the vision condition ($P = 0.013$). These results indicate that older adults produced greater force variability and lower force complexity than younger adults.

### Bilateral coordination: Vector coding and UCM analysis

The three-way mixed ANOVA on the in-phase frequency indicated significant two interactions: (A) Group × Force Level interaction [$F(1, 48) = 4.783$; $P = 0.034$; partial $\eta^2 = 0.091$; Fig 4a] and (B) Force Level × Vision Condition interaction [$F(1, 48) = 29.476$; $P < 0.001$; partial $\eta^2 = 0.380$]. However, no significant differences in in-phase frequency were found

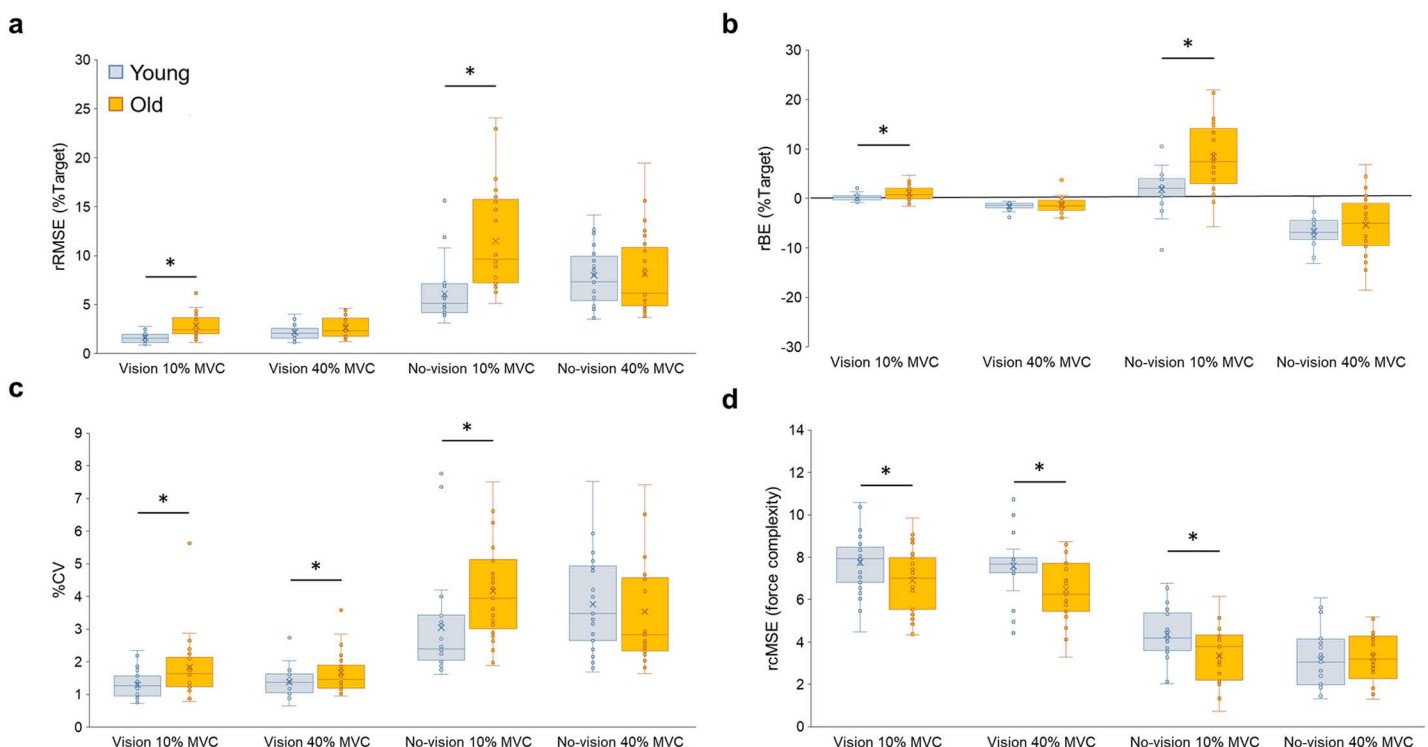

**Fig 3. Bilateral force control capabilities for the younger and older groups.** (a) rRMSE, (b) rBE, **(c)** %CV, and **(d)** MSE. Box plot shows individual data (circles), mean (X sign in the box), median (horizontal line in the box), interquartile range (IQR = Q3-Q1; top and bottom of the box indicates Q3, and Q1), maximum value: Q1 + 1.5 × IQR, and minimum value: Q1 – 1.5 × IQR. Asterisk (*) indicates a significant difference between groups (*P* < 0.05). Other significant findings based on vision and force level conditions were stated in S1 Table.

between younger and older groups (Fig 4a). The analysis of the anti-phase frequency showed significant two interactions: (A) Group × Force Level interaction [$F(1, 48) = 18.309$; $P < 0.001$; partial $\eta^2 = 0.276$; Fig 4b] and (B) Force Level × Vision Condition interaction [$F(1, 48) = 6.614$; $P < 0.013$; partial $\eta^2 = 0.121$]. Bonferroni's pairwise comparisons on Group × Force Level interaction findings reported that the older group revealed lower anti-phase frequency than the younger group at 40% of MVC collapsed across vision conditions ($P = 0.001$). Taken together, these results indicate that older adults used compensatory anti-phase coordination less frequently than younger adults.

The three-way mixed ANOVA on $V_{Index}$ revealed significant Group × Vision Condition interaction [$F(1, 48) = 4.863$; $P = 0.032$; partial $\eta^2 = 0.092$; Fig 4c]. In the vision condition, the older group showed lower $V_{Index}$ values than the younger group collapsed across force levels ($P < 0.001$). The analysis of good variability indicated a significant Force Level main effect [$F(1, 48) = 27.122$; $P < 0.001$; partial $\eta^2 = 0.361$], but there was no significant difference between the younger and older groups. Finally, the analysis of the bad variability showed Group main effect [$F(1, 48) = 4.117$; $P = 0.048$; partial $\eta^2 = 0.079$; Fig 4d] and Force Level × Vision Condition interaction [$F(1, 48) = 26.330$; $P < 0.001$; partial $\eta^2 = 0.354$]. The older group produced greater bad variability than the younger group collapsed across force levels and vision conditions. These findings indicated that older adults showed lesser synergy with greater bad variability between feet than younger adults across multiple trials.

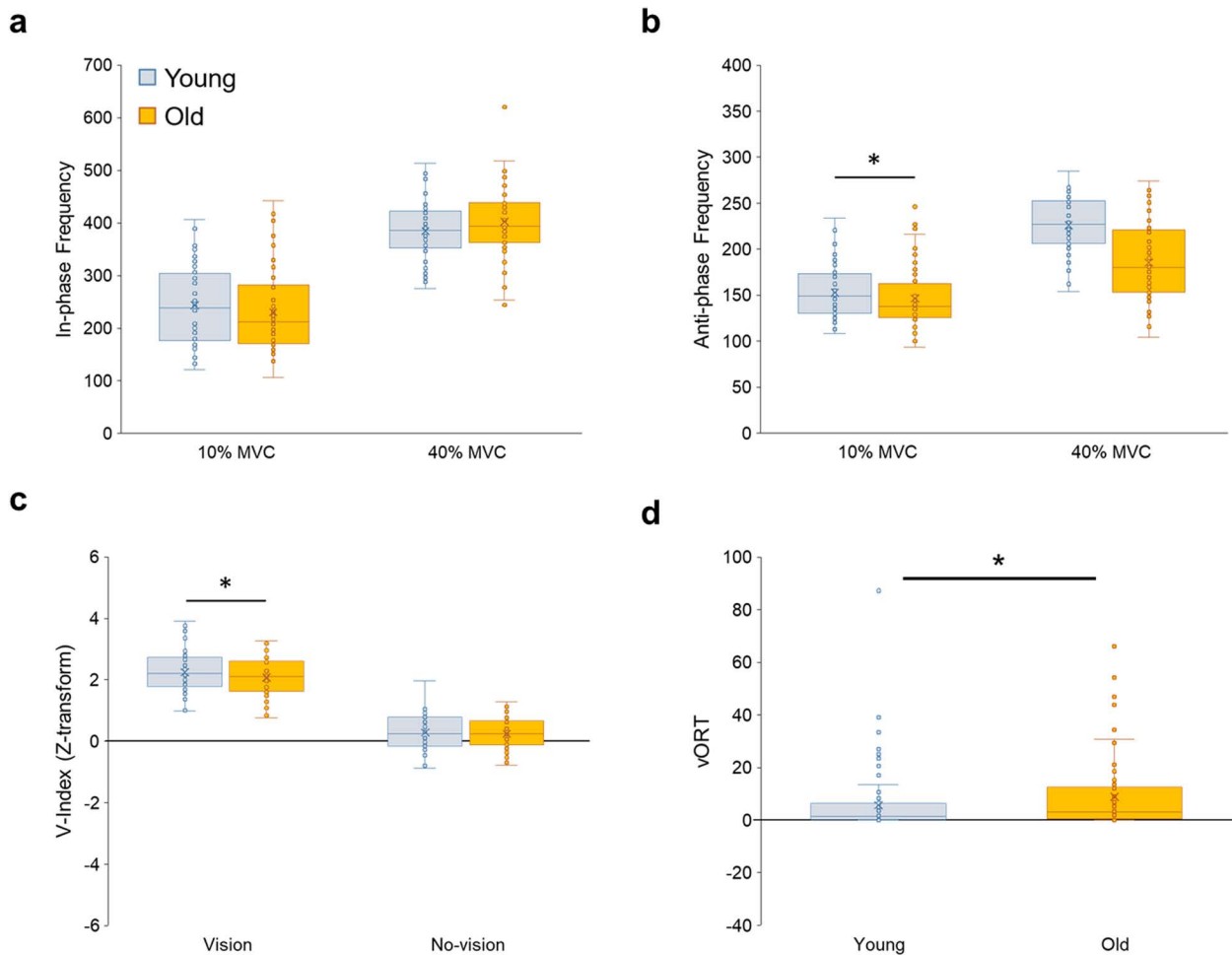

**Fig 4. Bilateral force coordination for the younger and older groups. (a)** In-phase frequency from vector coding analysis. **(b)** Anti-phase frequency from vector coding analysis. **(c)** Bilateral motor synergies ($V_{Index}$) from UCM analysis. **(d)** Bad variability ($V_{ORT}$) from UCM analysis. Box plot shows individual data (circles), mean (X sign in the box), median (horizontal line in the box), interquartile range (IQR = Q3-Q1; top and bottom of the box indicates Q3, and Q1), maximum value: Q1 + 1.5 × IQR, and minimum value: Q1 − 1.5 × IQR. Asterisk (*) indicates a significant difference between groups ($P < 0.05$). Other significant findings based on vision and force level conditions were stated in S2 Table.

### Correlation finding between force control variables and age in the older group

For the older group, Pearson's correlation analyses reported that significant correlation between force complexity and age (Fig 5 and S3 Table): (1) increased age versus lower rcMSE at 10% of MVC in the vision condition ($r = -0.432$; $P = 0.031$; Fig 5a) and (2) increased age versus lower rcMSE at 40% of MVC in the no-vision condition ($r = -0.409$; $P = 0.043$; Fig 5b). These results indicated that decreased force complexity was related to increased age for the older group.

## Discussion

This study investigated bilateral fine motor control capabilities of lower extremities in older people using an isometric ankle dorsiflexion force control paradigm. In comparison with younger adults, older adults had increased force error with a tendency to overshoot at 10%

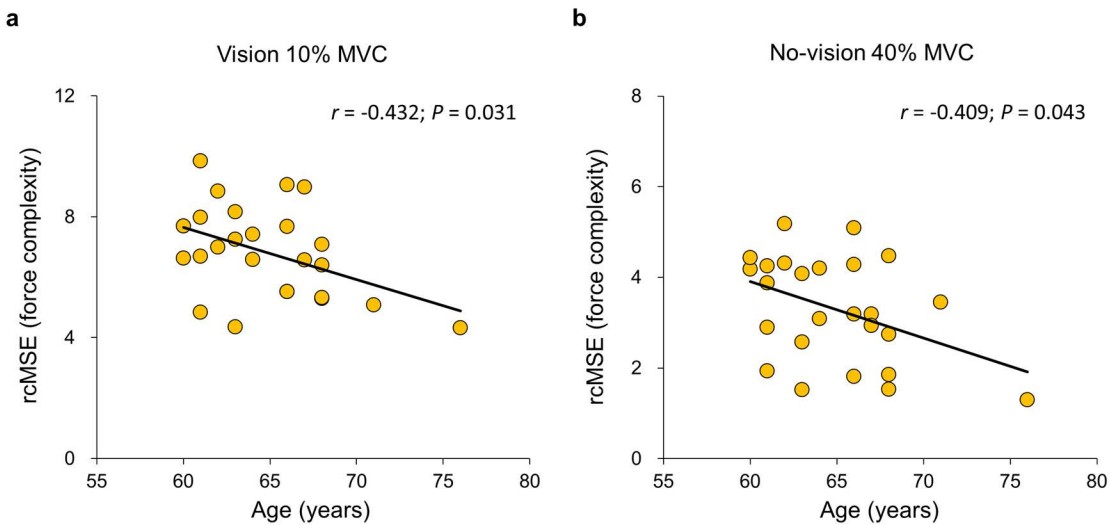

**Fig 5. Correlation between Age and force complexity in the older group. (a)** Correlation in the vision condition at 10% of MVC and **(b)** Correlation in the no-vision condition at 40% of MVC.

of MVC across vision and no-vision conditions. Moreover, bilateral ankle dorsiflexion forces produced by older adults were more variable and less complex, as indicated by conventional and nonlinear approaches to quantifying motor variability. Regarding interlimb force coordination strategies, older adults showed less anti-phase coordination within a trial and a reduction of motor synergies across multiple trials. Finally, the correlation analyses found that increased age was related to lower force complexity in older adults.

The lower accuracy of bilateral ankle dorsiflexion forces in older people expanded previous findings of increased bimanual hand-grip and wrist-extension force errors by demonstrating that aging may induce interlimb motor control deficits across the upper and lower extremities [32,49]. Spedden and colleagues reported that tracing error at 10% of MVC increased for older adults during unilateral ankle dorsiflexion force control task [20]. Further, the greater force tracing error was significantly associated with impaired bidirectional connectivity between the primary motor area (M1) and the premotor cortex (PMC) within the dominant hemisphere [20]. Further, previous findings indicated altered interhemispheric connectivity patterns (e.g., bilateral sides of the M1 and PMC) in older adults because of potential structural changes in the corpus callosum [50,51]. Specifically, higher neural connections in the bilateral motor network developed in older adults [52,53], and despite the unclear neural mechanisms underlying this relationship, certain patterns (e.g., resting state connectivity) were associated with poorer bimanual motor performances [54]. Although cumulative findings were limited to the upper extremity motor control tasks, altered neural communication between hemispheres with aging appears to influence bilateral motor control capabilities in the lower extremities. Importantly, the current study did not directly measure brain activation patterns between older and younger adults during bilateral ankle dorsiflexion force control tasks. Thus, future studies should focus on brain activation changes to examine potential neurophysiological mechanisms underlying age-related changes in bilateral ankle force control.

Moreover, the greater variability of bilateral ankle dorsiflexion forces at 10% and 40% of MVC support recent meta-analytic findings that age-induced loss of force steadiness was typically observed in smaller and distal muscle groups across various submaximal force levels [17]. In particular, our results confirmed that the increased variability of ankle dorsiflexion

forces in older adults emerge in bilateral conditions in addition to the unilateral condition [16,55,56]. Prior studies reported that aging may induce greater temporal and spatial variability of locomotion [40,57,58]. For example, older adults aged 60–86 years showed significant correlation between increased age and greater variability of gait variables (e.g., step length, width, and double support time) [57]. Further, increased gait variability may be associated with higher risk of falls [58,59]. Interestingly, significant correlation between gait variability and ankle dorsiflexion force variability at 10% of MVC was observed in the older group as well [40]. These findings suggested that more variable gait patterns as well as higher fall risk in older adults may be related to increased variability of bilateral ankle dorsiflexion forces. Potential mechanisms underlying excessive force variability with aging involved the motor unit reorganization. Aging may cause a loss of motor units leading to the remaining alpha motor neurons reinnervated with some of the denervated muscle fibers. Because aging muscles may have fewer and larger motor units, these larger motor units are recruited early even at lower targeted force levels, leading to more variable muscle forces [17,60,61]. This phenomenon may increase in distal muscle groups because of the greater effects of motor unit reorganization on smaller muscles [17,21,62]. Older adults have a smaller number of motor units and reduced excitability in motor neuron pools in the tibialis anterior [63–65]. Potentially, age-related motor unit remodeling may increase motor variability, presumably resulting in higher task errors with more overshooting patterns at the lower targeted level in this study.

The nonlinear method indicated bilateral motor control deficits in the lower extremities of older adults by identifying loss of complexity for bilateral ankle dorsiflexion forces across all vision conditions. In addition, a significant correlation was found between lower complexity of bilateral ankle dorsiflexion forces and increased age. According to the loss of complexity hypothesis [66], aging may interfere with the neuromuscular system, resulting in the progressive loss of complexity within the dynamics of physiological signals. Previous findings supported this assumption by demonstrating the lower complexity of isometric forces in small muscles of the upper extremities (e.g., index finger abduction force) at 5%–40% of MVC [67–70], and these patterns were also observed in the lower extremity muscles, such as the plantar flexors and knee extensors [71,72]. These findings posited that a loss of complexity in the aging appeared in both small and large muscle groups because of the increased common synaptic input, leading to impairments in fine motor control and gait performances [19,73,74]. To the best of our knowledge, this study is the first to demonstrate the loss of complexity in bilateral ankle dorsiflexion forces in older people and suggest that aging may progressively debilitate the ability of lower extremities to instantly adapt motor outputs in response to environmental changes. Decreased motor adaptability in bilateral ankle dorsiflexors may be responsible for loss of locomotion complexity in the aging population [75].

During bilateral ankle dorsiflexion force control tasks, older adults revealed less anti-phasic ankle dorsiflexion coordination within a single trial, as indicated by the decreased frequency in anti-phase ratio. These findings are consistent with previous results that older people showed more in-phase coordination patterns while modulating force production between hands [28,32]. These coordinative patterns in older adults typically represent less compensatory and adjustable motor actions between extremities [30,76]. The present findings support that older adults may exhibit impairments in interlimb force coordination across the upper and lower extremities in comparison with younger adults. Previously, impaired movement coordination between legs during locomotion, as indicated by less anti-phase coordination (i.e., lower accuracy of relative step timing between legs), was observed in older adults [77,78]. Moreover, a reduction of anti-phase coordination patterns may increase fall risk in older adults because these compensatory actions between legs are essential for maintaining balance and stability in response to altered environments [79,80]. Despite the lack of kinematic data

in this study, the potential relationship between bilateral kinetic and kinematic coordination patterns should be examined in the future studies because of the contribution of ankle joint force control capabilities to gross movement control, such as gait and postural control [81].

In the vision condition, older adults produced lower bilateral motor synergies between feet with greater bad variability indicating poor interlimb coordination across multiple trials. According to the UCM theory, increased bilateral motor synergies indicate greater stabilization of task performances, which are highly affected by changes in bad variability (i.e., the variance of fundamental elements projected to the undesired space). Previous studies have revealed that bilateral motor synergies in healthy young participants increased when they performed bilateral knee and ankle force control tasks that required more challenging goals (i.e., intentionally asymmetrical force productions between legs) [36,37]. These findings raised a possibility that people with more complex coordination such as anti-phasic interlimb force outputs within a trial would demonstrate greater synergetic interlimb motor actions across multiple trials. Like previous studies showing upper extremity deficits in age-related interlimb coordination (e.g., decreased anti-phase coordination and motor synergies during bimanual finger and wrist force control tasks) [32,82], our UCM findings also suggest that aging induces impaired lower limb coordination between trials as well as within a trial.

The current findings suggest that older adults may experience deficits in bilateral ankle dorsiflexion force production potentially associated with impaired locomotion and fall risk. Potentially, exoskeleton robots may be a viable option for improving bilateral ankle motor control because these approaches can provide accurate support and resistance to strengthen targeted muscles contributing to functional recovery of ankles [83,84]. For example, the application of ankle exoskeletons in older adults demonstrated positive effects as assistive devices for locomotion and effective rehabilitation tools for bilateral ankle functions [85,86]. Specifically, exoskeleton assistance on bilateral ankles reduced metabolic cost and improved gait speed in older adults [86,87]. A case study reported that four weeks of resistance training using bilateral ankle assistive robots increased left and right ankle muscle strength, gait speed, and soleus activations during gait for an older adult [85]. Thus, future studies are necessary to determine whether robot-based ankle rehabilitation programs can improve bilateral ankle dorsiflexion force control and interlimb coordination in older adults.

Importantly, the current findings have several potential limitations. First, a bilateral force control task of this study required participants to perform similar motor actions between feet (i.e., ankle dorsiflexion forces). Given that various activities of daily living often involve asymmetrical force production between feet, how older adults deal with these challenging force control tasks must be examined in future studies. Second, investigating the relationship between bilateral ankle dorsiflexion force control and movement control deficits in the lower extremities (e.g., a risk of fall) is necessary for the aging population. Third, impaired bilateral ankle dorsiflexion force control in the older group may be influenced by different weight or muscle mass that were not controlled for group comparisons. Finally, neuroimaging and motor unit analysis studies for motor control of the lower extremities are still insufficient. Thus, additional approaches that focus on bilateral lower limb force control paradigms while collecting brain and motor neuron activation data should be conducted for older adults.

## Conclusion

In conclusion, we found that older people had deficits in bilateral ankle dorsiflexion force control performances, including a reduction of force accuracy, increased force variability, and loss of complexity. Moreover, interlimb coordination patterns between feet for older adults were altered with less anti-phase coordination within a trial and decreased bilateral motor synergies between trials. Given that bilateral ankle dorsiflexion force control may be crucial for postural

control, walking, and climbing stairs [7–10], effective training programs are required to prevent fall risks and promote independent life in the aging population [88]. Potentially, advancing the central and peripheral nervous system using neuromodulation techniques such as transcranial direct current stimulation (tDCS) and functional electrical stimulation combined with movement training may be viable options for older adults. For example, anodal tDCS on either M1 or cerebellar regions (i.e., key regions for fine motor control and coordination) showed large effects on improved balance control capabilities in older adults [89]. Despite the limited number of studies that have examined the effects of neuromuscular electrical stimulation (NMES) on quadriceps muscles, the findings suggested potential improvements in muscle strength and mass in older people [90]. Thus, future studies may investigate effective protocols of tDCS and NMES specific for older adults to optimize their fine motor control capabilities of lower extremities.

## Supporting information

**S1 Table.  Bilateral force control capabilities for the younger and older groups.** MVC = maximum voluntary contraction, rRMSE = relative root mean square error, rBE- relative bias error, %CV = coefficient of variation, rcMSE = refined composite multiscale sample entropy. Data are mean ± SD. Number sign (#) means a significant difference between force levels. Ampersand sign (&) denotes a significant difference between vision conditions. ($P$ < 0.05). (DOCX)

**S2 Table.  Bilateral force coordination for the younger and older groups.** Data are mean ± SD. Number sign (#) means a significant difference between force levels. Ampersand sign (&) denotes a significant difference between vision conditions. ($P$ < 0.05). (DOCX)

**S3 Table.  Pearson's correlation between age and bilateral force control capabilities for the older group.** MVC = maximum voluntary contraction, rRMSE = relative root mean square error, rBE- relative bias error, %CV = coefficient of variation, rcMSE = refined composite multiscale sample entropy. Asterisk (*) indicates $P$ < 0.05. (DOCX)

**S1 Dataset.  Specific values of bilateral force control outcomes for each participant across force levels and vision conditions.** MVC = maximum voluntary contraction, rRMSE = relative root mean square error, rBE- relative bias error, %CV = coefficient of variation, rcMSE = refined composite multiscale sample entropy. (PDF)

## Acknowledgments

The authors sincerely thank you for the study participant.

## Author contributions

**Conceptualization:** Do-Kyung Ko, Nyeonju Kang.

**Data curation:** Do-Kyung Ko, Hanall Lee, Hajun Lee.

**Formal analysis:** Do-Kyung Ko.

**Funding acquisition:** Nyeonju Kang.

**Investigation:** Do-Kyung Ko, Nyeonju Kang.

**Methodology:** Nyeonju Kang.

**Project administration:** Hanall Lee, Nyeonju Kang.

**Resources:** Nyeonju Kang.

**Software:** Do-Kyung Ko, Nyeonju Kang.

**Supervision:** Nyeonju Kang.

**Validation:** Do-Kyung Ko, Nyeonju Kang.

**Visualization:** Do-Kyung Ko.

**Writing – original draft:** Do-Kyung Ko, Nyeonju Kang.

**Writing – review & editing:** Do-Kyung Ko, Hanall Lee, Hajun Lee, Nyeonju Kang.

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
