## [Decision Letter · Decision Letter 0]

30 Dec 2024

PONE-D-24-52865Bilateral ankle dorsiflexion force control impairments in older adultsPLOS ONE

Dear Dr. Kang,

Thank you for submitting your manuscript to PLOS ONE. After careful consideration, we feel that it has merit but does not fully meet PLOS ONE’s publication criteria as it currently stands. Therefore, we invite you to submit a revised version of the manuscript that addresses the points raised during the review process.

We look forward to receiving your revised manuscript.

Kind regards,

Tomoyoshi Komiyama, Ph.D

Academic Editor

PLOS ONE

Journal Requirements:

 “This work was supported by Incheon National University Research Grant in 2024 (2024-0059).”

“The authors report no declarations of interest.”

Additional Editor Comments:

Dear Authors,

Your study found that older adults had deficits in bilateral ankle dorsiflexion force control performance, and that interlimb coordination patterns between the feet were altered. You also found that the complexity of bilateral interfoot force control decreased with age. These results indicate that aging may impair the sensorimotor control ability of the lower limbs.

I reviewed the comments from the two referees and determined that your manuscript requires significant revisions. I thus decided to minor revision of this manuscript and ask you to resubmit after revising it according to these reviews.

I believe these comments will be very helpful in the revision of your study.

Tomoyoshi Komiyama

Reviewers' comments:

Reviewer's Responses to Questions

**Comments to the Author**

1. Is the manuscript technically sound, and do the data support the conclusions?

Reviewer #1: Yes

Reviewer #2: Yes

2. Has the statistical analysis been performed appropriately and rigorously? 

Reviewer #1: Yes

Reviewer #2: Yes

3. Have the authors made all data underlying the findings in their manuscript fully available?

Reviewer #1: Yes

Reviewer #2: Yes

4. Is the manuscript presented in an intelligible fashion and written in standard English?

Reviewer #1: Yes

Reviewer #2: Yes

5. Review Comments to the Author

Reviewer #1: Abstract

1. Consider a sentence to introduce why you want to investigate this muscle in particular.

2. Line 46-47: was this difference significant?

3. At the end of the abstract, consider a sentence regarding applicability and significance.

Overall the Abstract was well written, but could have additional sentences to justify why the study was conducted and what significance the results have.

Introduction

1. Good first paragraph identifying the need to examine the ankle dorsiflexors.

2. Lines 71-75: Probably don’t need to discuss progression of isometric ankle force control from childhood to adulthood, considering that is not investigated.

3. Line 95-98: Cite

4. May be good to identify specifically the novelty of your study, i.e. bilateral analysis. You mentioned others used unilateral, but would this be the first to investigate bilateral?

5. Was there a reason why you chose bilateral movement, as opposed to testing both legs independently?

Overall the introduction was well written. There is clear justification for the existing literature, the gaps in knowledge, and the purpose for the current study

Methods

1. Unsure if specific dates of recruitment are necessary.

2. Were subjects recruited to match by gender? What about physical activity matching? There was a significant starting difference in ankle dorsiflexion, could you make strength relative to subject weight to minimize difference in analysis?

3. How were subjects fitted to the testing device? You mention joint angles, how was this established for each individual?

4. Line 144: Did the subjects do any sort of familiarization trials?

5. Was the total MVC force value the combination of the two transducers?

6. Line 156: You mention ten consecutive trials for each block, 30 seconds between trials and 60 seconds between conditions. Was this all performed on the same day? Why did you choose these rest periods? Was the 4 blocks performed randomly? If not, do you think performance on last block may be impacted by fatigue?

Overall the methods included much of the necessary components. The researcher should consider providing more information about the specifics of recruitment, subject matching, randomization of blocks, and why the protocol they used was chosen regarding rest periods.

Results

Again, would be nice to know if groups were matched in some way other than gender.

Line 242 – unsure if you are saying older group had significantly higher RMSE across vision and no vision at 10%, or the “two vision” conditions (10 & 40%)

Line 246 – Again, the “two vision condition” doesn’t make sense

Line 246-247 – Should it be “older adults showed greater overshooting at 10% of MVC in the no-vision condition compared to the vision condition”

Figure 3 – The legend is a bit confusing and very long. May find a way to make it simpler to understand

Line 274 – Was this lower anti-phase in older group irrespective of condition at 40% or with both vision and no vision

Figure 4 – same, legend is long and confusing

Line 299 – what was negatively correlated with force complexity, age?

The results section is very in depth and a bit hard to read at times. It is hard to be sure/remember what each variable measured and could be better written to make known what the post hoc significance was. May read better with most result values presented in a table so less numbers in text. Overall, not bad but could be written to make it easier for the reader to comprehend what is going on.

Discussion

Line 316 – cite the figure this statement is referring to

Line 323-326 – Very long sentence, considered separating

Line 320-334 – While I appreciate this reference to previous work on the subject, I would make mention at the end of this paragraph that your study didn’t specifically measure anything regarding the motor cortex or any area of the brain.

Line 335 – 350 – May be a good spot to mention how the variability during bilateral tasks may have significance with impaired walking or fall susceptibility in older adults

Line 369-382 – Again, would be nice to have a sentence about how less anti-phase and less compensatory adjustments may lead to fall risk

Line 393 – 396 – Sentence is worded odd maybe try “Like previous studies showing upper extremity deficits in age-related interlimb coordination (28, 70), our UCM findings also suggest that aging induces impaired lower limb coordination between trials as well as within a trial.

For limitations, I think it is important that you make reference to lack of matching to physical activity, weight, or muscle mass. While I can appreciate that older groups will likely have less muscle and less physical activity, I think that is important to reference as well.

Overall, I greatly enjoyed reading this research and think this is important to add to the existing literature for older adults and help them develop better ways to rehabilitate and treat with therapeutic approaches.

Reviewer #2: The authors have presented an ankle plantar-dorsiflexion force control study. Several such studies have been reported in literature even by using ankle assistive robot exoskeletons. Can the authors please emphasize on the contribution of this study in the Introduction and Discussion/conclusion sections?

6. PLOS authors have the option to publish the peer review history of their article (what does this mean? ). If published, this will include your full peer review and any attached files.

**Do you want your identity to be public for this peer review?** For information about this choice, including consent withdrawal, please see our Privacy Policy .

Reviewer #1: No

Reviewer #2: No

---

## [Author Response · Author response to Decision Letter 1]

16 Jan 2025

Response to Reviewers

General: We appreciate the Reviewers’ positive comments. This process leads to a more comprehensive and vastly improved manuscript.

Reviewer #1

1. [Abstract] Consider a sentence to introduce why you want to investigate this muscle in particular.

Response: Thank you for the suggestion. We have added a sentence to the Abstract for clarifying why we focused on ankle dorsiflexion force control: Age-related impairments in ankle dorsiflexion force modulation are associated with gait and balance control deficits and greater fall risk in older adults.

2. [Abstract] Line 46-47: was this difference significant?

Response: Our apologies for the confusing sentence. All findings in the Abstract were confirmed with statistical significance (P < 0.05). We removed ‘tendency’ from the sentence and added ‘significantly’ to sentences.

3. [Abstract] At the end of the abstract, consider a sentence regarding applicability and significance.

Response: We have added a sentence to the Abstract: Considering the importance of ankle dorsiflexion for executing many activities of daily living, future studies may focus on developing training programs for advancing bilateral ankle dorsiflexion force control capabilities.

4. [Abstract] Overall the Abstract was well written, but could have additional sentences to justify why the study was conducted and what significance the results have.

Response: Consistent with the Reviewer’s suggestion, we revised the Abstract with new sentences emphasizing study significance and future directions. Thank you so much.

5. [Introduction] Good first paragraph identifying the need to examine the ankle dorsiflexors.

Response: We appreciate your positive comments.

6. [Introduction] Lines 71-75: Probably don’t need to discuss progression of isometric ankle force control from childhood to adulthood, considering that is not investigated.

Response: We agree. The sentence was revised by focusing on age-induced force control deficits: Given that age-related changes in the central and peripheral nervous systems may affect the ability to simultaneously correct and maintain isometric force outputs near a targeted level [1, 2], isometric ankle force control capabilities were impaired with aging [3-5].

7. [Introduction] Line 95-98: Cite

Response: Our apologies for missing references. We added relevant references: Interlimb force coordination strategies have two aspects to consider: (1) coordination pattern within-trial analysis and (2) coordination pattern between-trial analysis. A former approach can estimate the strength of interlimb force coordination according to in-phase and anti-phase actions between feet so that more anti-phasic force coordination within a trial predominantly appeared with better bilateral force control performances [6, 7]. A latter approach can assess how a performer successfully coordinates interlimb motor actions synergistically across repetitive trials [8, 9].

8. [Introduction] May be good to identify specifically the novelty of your study, i.e. bilateral analysis. You mentioned others used unilateral, but would this be the first to investigate bilateral?

Response: Thank you for the suggestion. To the best of our knowledge, this is the first study to investigate bilateral ankle dorsiflexion force control capabilities in aging population. We added this information to the Introduction.

9. [Introduction] Was there a reason why you chose bilateral movement, as opposed to testing both legs independently?

Response: In fact, age-induced movement control deficits were frequently observed in walking, postural control, and sit-to-stand that require bilateral actions in the lower extremities [10]. Given that many daily activities for living independent life consist of fundamental motor skills related to interlimb coordination between feet [11, 12], investigating bilateral force control and coordination in the lower extremities is necessary for understanding age-induced functional impairments. Studies on bilateral ankle dorsiflexion force control capabilities in older adults may provide additional information on how aging influences cooperative fine motor control between feet. Further, these findings can be used for developing new rehabilitation programs (e.g., robotic exoskeletons for improving lower limb coordination) targeting functional recovery of bilateral ankle movements in aging population [13]. We added this information to the Introduction. Thank you.

10. [Introduction] Overall the introduction was well written. There is clear justification for the existing literature, the gaps in knowledge, and the purpose for the current study

Response: We appreciate your positive comments. After revision based on the Reviewer’s suggestion, the clarity of Introduction increased. Thank you.

11. [Methods] Unsure if specific dates of recruitment are necessary.

Response: We added the recruitment period in the Methods because the Journal requires the information based on the Human Participants Research Checklist.

12. [Methods] Were subjects recruited to match by gender? What about physical activity matching? There was a significant starting difference in ankle dorsiflexion, could you make strength relative to subject weight to minimize difference in analysis?

Response: For both groups, the gender ratio (female: male = 15:10) and physical activity level were matched. We added this information to the Methods and Table 1.

However, demographic characteristics (e.g., weight) can affect starting difference in ankle dorsiflexion force within a group as well as between groups. To minimize effects of different ankle dorsiflexion strength across participants on force control tasks, we normalized targeted force levels based on each participant's maximum voluntary contraction (MVC) including 10% and 40% of MVC [14]. In addition, we used relative outcome variables on bilateral force control performances (i.e., relative root-mean-square error, relative bias error, and coefficient of variation) to minimize potential distortions by individual force levels [15]. We added the information to the Methods. Thank you.

13. [Methods] How were subjects fitted to the testing device? You mention joint angles, how was this established for each individual?

Response: We directly adjusted the chair and instructed them to maintain proper joint positions following joint angle ranges: 90°–95° of hip flexion, 90°–100° of knee flexion, and approximately 90° of ankle dorsiflexion (Fig. 1a). During tasks, we continuously monitored and ensured that participants maintain proper joint positions. We added this information to the Methods.

14. [Methods] Line 144: Did the subjects do any sort of familiarization trials?

Response: Yes, before starting an experimental block, we provided one practice trial for familiarization. We added this information to the Methods.

15. [Methods] Was the total MVC force value the combination of the two transducers?

Response: Yes, individual’s MVC force value indicates maximal value of the sum of the left and right foot forces.

16. [Methods] Line 156: You mention ten consecutive trials for each block, 30 seconds between trials and 60 seconds between conditions. Was this all performed on the same day? Why did you choose these rest periods? Was the 4 blocks performed randomly? If not, do you think performance on last block may be impacted by fatigue?

Response: We administered four experimental blocks on the same day for each participant. To minimize potential fatigue and practice effects on bilateral force control performances, we randomly provided four experimental blocks. Further, we ensured enough rest time for preventing fatigue effects based on previous studies [6, 16, 17]. We added references regarding rest periods in bilateral ankle dorsiflexion force control tasks.

17. [Methods] Overall the methods included much of the necessary components. The researcher should consider providing more information about the specifics of recruitment, subject matching, randomization of blocks, and why the protocol they used was chosen regarding rest periods.

Response: Thank you for your positive comments. Consistent with the Reviewer’s suggestion, we provided specific information on participant and experimental procedures.

18. [Results] Again, would be nice to know if groups were matched in some way other than gender.

Response: Yes, we confirmed that the gender ratio and physical activity level were matched between older and younger groups. We added this information in the Participants section of Methods.

19. [Results] Line 242 – unsure if you are saying older group had significantly higher RMSE across vision and no vision at 10%, or the “two vision” conditions (10 & 40%)

Response: Our apologies for confusing expression. We revised the sentences and added P-value from the post-hoc analysis for each significant finding: Bonferroni’s pairwise comparisons revealed that the older group produced higher rRMSE values than the younger group at 10% of MVC for vision (P < 0.001) and no-vision conditions (P < 0.001), respectively.

20. [Results] Line 246 – Again, the “two vision condition” doesn’t make sense.

Response: Our apologies for confusing expression. We revised the sentences and added P-value from the post-hoc analysis for each significant finding: In the post hoc analyses, the older group showed greater overshooting than the younger group at 10% of MVC for vision (P = 0.007) and no-vision conditions (P < 0.001), respectively.

21. [Results] Line 246-247 – Should it be “older adults showed greater overshooting at 10% of MVC in the no-vision condition compared to the vision condition”

Response: We removed the sentence, and revised prior sentence based on the Reviewer’s suggestion. Please see above answer to the Reviewer’s question #20. Thank you so much.

22. [Results] Figure 3 – The legend is a bit confusing and very long. May find a way to make it simpler to understand

Response: Our apologies for confusing expression. We revised the figure legend and figures mainly focusing on group differences. Instead, we added supplementary table (S1 Table) to report other significant findings based on vision and force level conditions. Thank you.

23. [Results] Line 274 – Was this lower anti-phase in older group irrespective of condition at 40% or with both vision and no vision

Response: We revised the sentence: Bonferroni’s pairwise comparisons on Group × Force Level interaction findings reported that the older group revealed lower anti-phase frequency than the younger group at 40% of MVC collapsed across vision conditions (P = 0.001).

24. [Results] Figure 4 – same, legend is long and confusing

Response: Our apologies for confusing expression. We revised the figure legend and figures mainly focusing on group differences. Instead, we added supplementary table (S2 Table) to report other significant findings based on vision and force level conditions. Thank you.

25. [Results] Line 299 – what was negatively correlated with force complexity, age?

Response: We removed the confusing term and revised the paragraph: For the older group, Pearson’s correlation analyses reported that significant correlation between force complexity and age (Fig 5 and S3 Table): (1) increased age versus lower rcMSE at 10% of MVC in the vision condition (r = −0.432; P = 0.031; Fig 5a) and (2) increased age versus lower rcMSE at 40% of MVC in the no-vision condition (r = −0.409; P = 0.043; Fig 5b). These results indicated that decreased force complexity was related to increased age for the older group.

26. [Results] The results section is very in depth and a bit hard to read at times. It is hard to be sure/remember what each variable measured and could be better written to make known what the post hoc significance was. May read better with most result values presented in a table so less numbers in text. Overall, not bad but could be written to make it easier for the reader to comprehend what is going on.

Response: Consistent with the Reviewer’s suggestions, we extensively revised the Results. We added clear expression to state post hoc results and revised figure legends as well. Further, we tried to focus on group differences of variables and added other significant findings in supplementary tables. We think that the clarity of Results increased after revision procedures. Thank you so much for your valuable comments.

27. [Discussion] Line 316 – cite the figure this statement is referring to

Response: Our apologies for confusing expression. The statement in original version of manuscript described our correlation findings (Fig 5). To increase the clarity of statement, we revised the sentence: The correlation analyses found that increased age was related to lower force complexity in older adults.

28. [Discussion] Line 323-326 – Very long sentence, considered separating

Response: Thank you for the suggestion. We separated the sentence: Spedden and colleagues reported that tracing error at 10% of MVC increased for older adults during unilateral ankle dorsiflexion force control task [4]. Further, the greater force tracing error was significantly associated with impaired bidirectional connectivity between the primary motor area (M1) and the premotor cortex (PMC) within the dominant hemisphere [4].

29. [Discussion] Line 320-334 – While I appreciate this reference to previous work on the subject, I would make mention at the end of this paragraph that your study didn’t specifically measure anything regarding the motor cortex or any area of the brain.

Response: Thank you for your suggestion. We have added sentences at the end of paragraph: Importantly, the current study did not directly measure brain activation patterns between older and younger adults during bilateral ankle dorsiflexion force control tasks. Thus, future studies should focus on brain activation changes to examine potential neurophysiological mechanisms underlying age-related changes in bilateral ankle force control.

30. [Discussion] Line 335 – 350 – May be a good spot to mention how the variability during bilateral tasks may have significance with impaired walking or fall susceptibility in older adults

Response: Thank you for the insightful suggestion. We added sentences regarding potential relationship between bilateral force variability and gait variability as well as fall risk: Prior studies reported that aging may induce greater temporal and spatial variability of locomotion [17-19]. For example, older adults aged 60–86 years showed significant correlation between increased age and greater variability of gait variables (e.g., step length, width, and double support time) [18]. Further, increased gait variability may be associated with higher risk of falls [19, 20]. Interestingly, significant correlation between gait variability and ankle dorsiflexion force variability at 10% of MVC was observed in the older group as well [17]. These findings suggested that more variable gait patterns as well as higher fall risk in older adults may be related to increased variability of variability of bilateral ankle dorsiflexion forces.

31. [Discussion] Line 369-382 – Again, would be nice to have a sentence about how less anti-phase and less compensatory adjustments may lead to fall risk

Response: Thank you for the suggestion. We added more sentences to explain how less anti-phase coordination may increase fall risk: Previously, impaired movement coordination between legs during locomotion, as indicated by less anti-phase coordination (i.e., lower accuracy of relative step timing between legs), was observed in older adults [21, 22]. Moreover, a reduction of anti-phase coordination patterns may increase fall risk in older adults because these compensatory actions between legs are essential for maintaining balance and stability in response to altered environments [23, 24].

32. [Discussion] Line 393 – 396 – Sentence is worded odd maybe try “Like previous studies showing upper extremity deficits in age-related interlimb coordination (28, 70), our UCM findings also suggest that aging induces impaired lower limb coordination between trials as well as within a trial.

Response: We re

---

## [Decision Letter · Decision Letter 1]

5 Feb 2025

Bilateral ankle dorsiflexion force control impairments in older adults

PONE-D-24-52865R1

Dear Dr. Kang,

We’re pleased to inform you that your manuscript has been judged scientifically suitable for publication and will be formally accepted for publication once it meets all outstanding technical requirements.

Kind regards,

Tomoyoshi Komiyama, Ph.D

Academic Editor

PLOS ONE

Additional Editor Comments (optional):

Dear authors,

Thank you for submitting your revised manuscript.

It was much easier to understand than the original manuscript.

I am satisfied with the responses and the edits, so I am happy to accept your study.

You have satisfactorily addressed the comments from the two reviewers.

Therefore, I have no further suggestions.

I believe this manuscript will satiate the reader's interest.

Tomoyoshi Komiyama

Reviewers' comments:

Reviewer's Responses to Questions

**Comments to the Author**

1. If the authors have adequately addressed your comments raised in a previous round of review and you feel that this manuscript is now acceptable for publication, you may indicate that here to bypass the “Comments to the Author” section, enter your conflict of interest statement in the “Confidential to Editor” section, and submit your "Accept" recommendation.

Reviewer #1: All comments have been addressed

Reviewer #2: All comments have been addressed

2. Is the manuscript technically sound, and do the data support the conclusions?

Reviewer #1: Yes

Reviewer #2: Yes

3. Has the statistical analysis been performed appropriately and rigorously? 

Reviewer #1: Yes

Reviewer #2: Yes

4. Have the authors made all data underlying the findings in their manuscript fully available?

Reviewer #1: Yes

Reviewer #2: Yes

5. Is the manuscript presented in an intelligible fashion and written in standard English?

Reviewer #1: Yes

Reviewer #2: Yes

6. Review Comments to the Author

Reviewer #1: Abstract: All revisions previously suggested were addressed adequately.

Introduction: All revisions previously suggested were addressed adequately.

Methods: All revisions previously suggested were addressed adequately. I appreciated your commitment to adding information about subject recruitment.

Results: All revisions previously suggested were addressed adequately. This section was confusing to read with the first submission but reads much better after revisions.

Discussion: All revisions previously suggested were addressed adequately.

I appreciate the authors work revising the paper to provide better information about the studies methods, and including certain information previously not within the first submission . Very good job.

Reviewer #2: Thank you for addressing my comments in the revised manuscript. The manuscript can be accepted for publication in its current form.

7. PLOS authors have the option to publish the peer review history of their article (what does this mean? ). If published, this will include your full peer review and any attached files.

**Do you want your identity to be public for this peer review?** For information about this choice, including consent withdrawal, please see our Privacy Policy .

Reviewer #1: No

Reviewer #2: No

---

## [Editor Report · Acceptance letter]

PONE-D-24-52865R1

PLOS ONE

Dear Dr. Kang,

I'm pleased to inform you that your manuscript has been deemed suitable for publication in PLOS ONE. Congratulations! Your manuscript is now being handed over to our production team.

Kind regards,

on behalf of

Dr. Tomoyoshi Komiyama

Academic Editor

PLOS ONE